# Rationale and population-based prospective cohort protocol for the disadvantaged populations at risk of decline in eGFR (CO-DEGREE)

Marvin Gonzalez-Quiroz,[1,2,3] Dorothea Nitsch,[3] Sophie Hamilton,[4] Cristina O'Callaghan Gordo,[5,6,7] Rajiv Saran,[8,9] Jason Glaser,[10] Ricardo Correa-Rotter,[11] Kristina Jakobsson,[12,13] Ajay Singh,[14] Nalika Gunawardena,[15] Adeera Levin,[16] Giuseppe Remuzzi,[17] Ben Caplin,[2] Neil Pearce,[18,19] on behalf of the DEGREE Study Steering Committee

BC and NP contributed equally.

For numbered affiliations see end of article.

**Correspondence to**
Dr Marvin Gonzalez-Quiroz;
m.quiroz@ucl.ac.uk

## ABSTRACT

**Introduction** A recently recognised form of chronic kidney disease (CKD) of unknown origin (CKDu) is afflicting communities, mostly in rural areas in several regions of the world. Prevalence studies are being conducted in a number of countries, using a standardised protocol, to estimate the distribution of estimated glomerular filtration rate (eGFR), and thus identify communities with a high prevalence of reduced glomerular filtration rate (GFR). In this paper, we propose a standardised minimum protocol for cohort studies in high-risk communities aimed at investigating the incidence of, and risk factors for, early kidney dysfunction.

**Methods and analysis** This generic cohort protocol provides the information to establish a prospective population-based cohort study in low-income settings with a high prevalence of CKDu. This involves a baseline survey that included key elements from the DEGREE survey (eg, using the previously published DEGREE methodology) of a population-representative sample, and subsequent follow-up visits in young adults (without a pre-existing diagnosis of CKD (eGFR<60 mL/min/1.73m$^2$), proteinuria or risk factors for CKD at baseline) over several years. Each visit involves a core questionnaire, and collection and storage of biological samples. Local capacity to measure serum creatinine will be required so that immediate feedback on kidney function can be provided to participants. After completion of follow-up, repeat measures of creatinine should be conducted in a central laboratory, using reference standards traceable to isotope dilution mass spectrometry (IDMS) quality control material to quantify the main outcome of eGFR decline over time, alongside a description of the early evolution of disease and risk factors for eGFR decline.

**Ethics and dissemination** Ethical approval will be obtained by local researchers, and participants will provide informed consent before the study commences. Participants will typically receive feedback and advice on their laboratory results, and referral to a local health system where appropriate.

## Strengths and limitations of this study

► We propose a prospective generic cohort protocol for populations affected by chronic kidney disease of unknown origin in which the sampling frame consists of the entire at-risk population. In addition, the use of this standardised protocol will allow for regional and international comparisons.

► Serial estimated glomerular filtration rate measurements in an apparently healthy population will allow the description of the evolution of disease and reduce problems associated with recall bias and reverse causation when assessing potential risk factors.

► Samples will be analysed in a single batch at the end of the study to minimise time-dependent measurement errors.

► A biobank is expected to be created in each centre to store biological samples for future analyses.

► As for any cohort, loss to follow-up could pose a threat to validity of the study, and every effort must be made to mitigate this.

## INTRODUCTION

A mysterious form of chronic kidney disease (CKD) is afflicting young adults, mostly in rural communities in a number of low-income and middle-income countries (LMICs).[1–10] This disease has been termed CKD of undetermined cause (CKDu). Several definitions for CKDu exist; the criteria typically include demonstration of renal damage using biomarkers in the absence of diabetes, severe hypertension or evidence of alternative renal diagnoses.[11–14] This syndrome has caused thousands of deaths and reduced the life expectancy among young adults in Mesoamerica, South Asia and possibly in other tropical/subtropical regions of the world.[7 15–19] The cause(s)

of CKDu are not yet established, but proposed aetiologies include recurrent dehydration/heat stress, pesticides, infections and heavy metals.[1] [20–22] In addition, there is no evidence that these forms of CKDu have a unified causality or are due to different aetiologies in diverse parts of the world.

Although a broad range of cross-sectional studies investigating prevalence of CKDu have been conducted in Mesoamerica, South Asia and other regions of the world,[1–7] [9] [17] these have generally not used standardised methodology, and therefore do not allow for valid international comparisons. A recently published standardised protocol (the Disadvantaged Populations eGFR Epidemiology Study (DEGREE) protocol) for estimating the population distribution of glomerular filtration rate (eGFR) has addressed this concern, and is being used in communities suspected to have a high prevalence of reduced eGFR. The DEGREE protocol makes it possible to undertake comparisons internationally, by mandating a population-representative sample and standardised collection of information on sociodemographic factors, occupational and environmental exposures, body composition and kidney function.[23] To date, studies using the DEGREE methodology have been conducted in four countries (Peru, Sri Lanka, India and Malawi), with a number of future projects in preparation or in progress.[17]

A recent meta-analysis highlighted the lack of robust studies that have considered risk factors for early kidney damage in CKDu.[24] This is of key importance as those with even apparently mildly damaged kidneys (eg, a borderline elevated serum creatinine (sCr) but no renal reserve) may experience progressive renal decline in response to a wide range of exacerbating insults (eg, episodes of dehydration/heat stress, nephrotoxic medication or other nephrotoxic exposures) making identification of causal associations challenging in those with existing kidney damage. Based on our experience,[25] [26] we propose a generic cohort protocol to characterise the decline in kidney function over time and conduct aetiological research in those without pre-existing CKD/risk factors at baseline but at risk of CKDu. Our focus is on conducting such cohort studies in populations that are at high risk for CKDu, that is, that have previously been classified as such by surveys based on cross-sectional eGFR measurements. In general, this work would follow on from a study using the DEGREE protocol, and hence we will use the term 'CO-DEGREE' (cohorts based on the DEGREE study) for such studies. Indeed, in some situations, a DEGREE survey may form the 'baseline', with a subgroup of DEGREE survey participants then being selected for follow-up based on age, a single measurement of eGFR ≥60 mL/min/1.73 m² (accepting that this is likely a conservative cut-off for pre-existing kidney dysfunction) and without clinical diagnosis or history of hypertension, diabetes mellitus, obesity or other known risk factor that could potentially explain CKD. However, the standardised protocol we propose here can also be used as a 'stand-alone' study design in any well-defined study group, without requiring that a DEGREE survey is conducted first.

We are already conducting such a cohort study in Nicaragua,[25] [26] and have had many challenges to address, including (1) community engagement, awareness of conditions, political unrest and ethics; (2) follow-up over time (frequency and minimising loss to follow-up); (3) fieldwork and laboratory standards to ensure decline is detected and (4) regular feedback information on study progress. We will draw on our experience in Nicaragua in presenting both the generic CO-DEGREE protocol and observations on the practical issues involved in conducting such studies in a particular population.

## OBJECTIVES

Studies using this generic cohort protocol, and contributing to the wider DEGREE collaboration, will aim to

1. Investigate the evolution of, and risk factors for, kidney function decline over time among populations at risk of CKDu.
2. Compare the evolution, and risk factors for kidney function decline, in different populations and regions at risk of CKDu.
3. Establish a framework for international collaboration and promote a network for future work on the causality of CKDu.

### Rationale for a cohort study of decline in eGFR
#### A representative sample of those at risk

Population-based cohort studies have several advantages:[27] first, this type of study allows the recruitment of a representative sample of the at-risk population, for example, it will include workers from a variety of occupations (including unemployed) at the community level. Assuming that the study sample is randomly selected from the entire at-risk population based on a community census, and there are no substantial problems with non-response, these studies are unlikely to be affected by significant selection bias. Furthermore, in contrast to studies conducted solely in an occupational setting, differential loss to follow-up is likely to be less problematic, particularly if workers are screened for kidney disease within that setting and potentially denied further work.

Like all prospective cohort studies, to ensure the entire population is 'at risk', those with the outcome at baseline should be excluded, although it is recognised that investigators may wish to follow up those with eGFR <60 mL/min/1.73 m² and those with established risk factors for CKD for other purposes (see below).

One general disadvantage of population-based studies is that this approach typically requires large sample sizes and long-term follow-up if disease is not highly prevalent. However, the focus of CO-DEGREE is on conducting studies in population with a high prevalence of CKDu (see below).[25] [27]

### Handling reverse causation and recall bias

The problem of reverse causation (eg, modification of behaviour or work tasks in response to the diagnosis of renal impairment) can be minimised in a cohort study by focusing on people without pre-existing disease, and then following these initially apparently 'healthy' participants over time. Similarly, a cohort approach, unlike cross-sectional studies, is less prone to recall bias regarding previous exposures.

### Measuring kidney function

Quantification of kidney function is most easily undertaken by determining sCr concentration, which is relatively easy and cheap to measure, and then calculating the eGFR. A case of CKDu is typically defined by an eGFR $<60\,mL/min/1.73\,m^2$ (sustained for at least 3 months to confirm chronicity) in the absence of known causes of kidney disease. However, this dichotomous definition has weaknesses in studies exploring the causation of CKDu, as it is well established that substantial damage may have already occurred at the histological level before serum biomarkers of renal dysfunction become abnormal (and other markers such as proteinuria are often absent in this disease). Furthermore, repeat measures after 3 months are not always performed in cross-sectional surveys, and sCr levels are modified by multiple non-renal factors such as high animal protein intake, strenuous exercise, changes in plasma volume, body mass index, sex, age, ethnicity and some drugs[28]; thus, cross-sectional studies examining associations with reduced eGFR based on a single sCr measurement may be prone to a significant degree of misclassification, especially in smaller studies.

Notably, the accuracy of sCr determinations is also an inherent problem (see further below). In addition, the Chronic Kidney Disease Epidemiology Collaboration (CKD-EPI) or The Modification of Diet in Renal Disease Study (MDRD) equations used to calculate eGFR from sCr[28] have not been validated in many populations reported to be suffering CKDu,[29] potentially further increasing misclassification bias in cross-sectional studies.

Alternative approaches based on serial eGFR measurements in the same person over time render between-person variation less problematic. If estimated across a period of time using multiple measures with sustained preanalytical and analytical quality, this will also reduce the influence of the within-person factors that are not directly related to kidney damage. In summary, an approach using serial eGFR measures substantially improves the potential to identify risk/causal factors for CKDu as well as allowing the description of the evolution of disease.

## CORE PROTOCOL
### Study design

This is a prospective cohort study protocol for studying decline in kidney function over time in populations with high reported prevalence of CKDu, primarily in LMICs. We consider the following study design issues: (1) population sampling strategy and follow-up interval, (2) questionnaire development and delivery, (3) clinical measurements and biosampling and (4) data management and reporting[25] (see figure 1). In addition, we

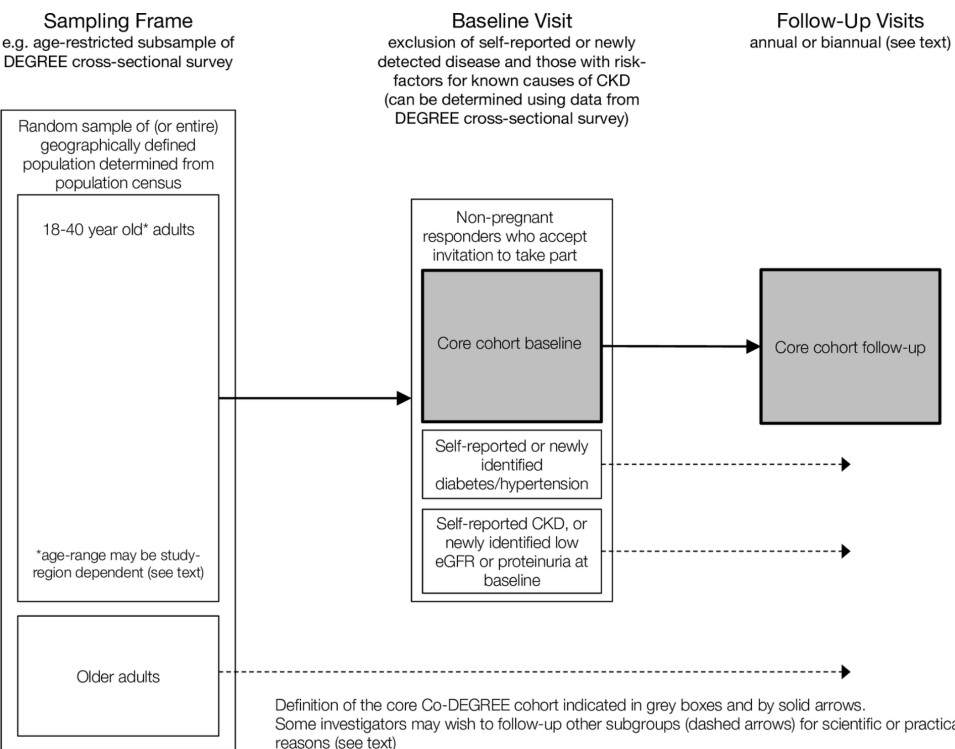

**Figure 1** Flowchart and study procedures of CO-DEGREE protocol.

discuss (1) sample size and follow-up duration and (2) ethical considerations.

## Population, sampling strategy and follow-up interval

In Mesoamerica, CKDu typically affects young men on the Pacific Coast. This population is dying in their 40s, often younger, from end-stage renal disease.[15 30] The disease appears to occur at a later age in South Asia, with few cases occurring in men in their 20s.[7 31] Nevertheless, one might expect preliminary changes in GFR to occur early in adulthood. In general, the study population should include participants who are old enough to experience an identifiable decline in kidney function, but not older age groups (eg, >60 years old) where the prevalence of CKD is already high in many populations globally (eg, up to 10%). Thus, inclusion criteria should be tailored to the local disease profile, but the default approach should be to recruit participants aged 18–40 years old (though 18–30 might be more appropriate in Central America, and 18–50 may be more appropriate in areas such as South Asia where age of onset appears older). The rationale for including people ≥18 years old was based on definition on adult life, and may be lowered, especially in populations where the working life starts years earlier. A population-census should be conducted to identify all potential participants in the appropriate age range and either the entire population recruited, or a random sample selected. In either case, response rates by age and sex should be reported.

The focus of these studies is to conduct aetiological research in those without traditional CKD/risk factors at baseline; thus, the sample size estimates (see below) are based on following a cohort in which those with evidence of pre-existing CKD, diabetes or hypertension have been excluded.[25] Diabetes can be diagnosed by self-report, use of medication or lab tests (fasting serum glucose: ≥7.0 mmol/L or HbA1C ≥48 mmol/mol),[32 33] and hypertension by self-report, use of medication or measurement

(seated, average BP ≥140/90 mmHg on second and third of three readings).[34]

In addition to self-report of CKD, those with previously detected eGFR <60 mL/min/1.73 m² , proteinuria (eg, albumin/creatinine ratio (ACR) >300 mg/g or dipstick 3+ or greater)[35] on testing at baseline should be excluded from the study. It is recognised that a proportion of participants not excluded by these criteria may still have some form of underlying kidney abnormality (eg, low-level proteinuria), and some of those excluded due to a low eGFR at baseline may go on to recover function, but this represents a pragmatic approach to excluding those with significant pre-existing renal disease at baseline. Furthermore, for practical, ethical or scientific reasons (eg, to gain insight into progression of established CKDu or other non-communicable disease research aims), investigators may wish to study an entire population (including those with pre-existing clinical diagnosis of, or newly identified, CKD, diabetes mellitus and hypertension), but in that case, it is important to ensure that there are sufficient 'disease-free' participants included at baseline to meet the sample size requirements (see table 1). Although the disease is generally more common in men, women with CKDu are of strong scientific interest in that they may suggest alternative risk factors, or help to rule out some that have been previously proposed. Hence, recruitment should in general involve equal numbers of men and women, though women who are pregnant at recruitment are also excluded, since pregnancy-related changes in eGFR are challenging to interpret.

The baseline study visit will require the administration of the core questionnaire, with additional context-specific additions, clinical measurements and biological samples. Subsequent to the baseline visit, follow-up visits should be conducted at least annually for a minimum follow-up of 2 years to evaluate the study outcome and keep close contact with the participants and update their

| Parameters | Scenario 1 | Scenario 2 | Scenario 3 | Scenario 4 | Scenario 5 | Scenario 6 | Scenario 7 | Scenario 8 |
|---|---|---|---|---|---|---|---|---|
| Population frequency of eGFR decline | 0.04 | 0.06 | 0.08 | 0.10 | 0.04 | 0.06 | 0.08 | 0.10 |
| Proportion population exposed | 0.5 | | | | | | | |
| OR associated with exposure | 2 | | | | 3 | | | |
| P (outcome\|unexposed) | 0.027 | 0.04 | 0.053 | 0.066 | 0.02 | 0.03 | 0.04 | 0.05 |
| P (outcome\|exposed) | 0.054 | 0.08 | 0.106 | 0.132 | 0.06 | 0.09 | 0.12 | 0.15 |
| Group size | 993 | 686 | 405 | 436 | 463 | 317 | 243 | 200 |
| Sample size | **1986** | **1372** | **810** | **872** | **926** | **634** | **486** | **400** |

**Table 1** Sample size calculations

Calculations based on equal proportion of the population exposed/unexposed for simplicity. No adjustments made for loss to follow-up or multiple testing.

P indicates probability, assumes 1-β=0.80; =0.05.

eGFR, estimated glomerular filtration rate.

**Table 2** Details and procedures of the baseline study visit and subsequence follow-up

| Items | Baseline visit (0 month) | Follow-up period (variable) | | | | At completion |
| --- | --- | --- | --- | --- | --- | --- |
| | | 12 months | 24 months | 36 months | 48 months | |
| Community census | X | – | – | – | – | – |
| Participants enrolment | X | – | – | – | – | – |
| Informed consent | X | | | | | |
| Update personnel contact information | X | X | X | X | X | |
| Anthropometric measurements | X | X | X | X | X | |
| Biological samples | X | X | X | X | X | |
| Baseline core questionnaire | X | – | – | – | – | |
| Follow-up questionnaire | | X | X | X | X | |
| Local serum creatinine measurement | X | X | X | X | X | |
| Results feedback | X | X | X | X | X | |
| Biobank | X | X | X | X | X | |
| Batch testing of serum creatinine | | | | | | X |

contact information. This will help minimise the loss to follow-up at each study point. Substantial seasonal variation in eGFR has been reported in a number of settings (both CKDu related and unrelated).[26 36–38] Therefore, the conduct of additional study visits at a 6-monthly interval (eg, at beginning and end of summer season) might be useful in explaining within-person eGFR variation as well as providing important information for the wider population on the significance of kidney function testing at different time points in the year (perhaps for a subset of participants or a proportion of the follow-up period) (see table 2).

## Questionnaires

The purpose of the baseline core questionnaire is to obtain a minimum dataset to explore associations with decline in kidney function and make comparisons within and between persons. The baseline core questionnaire (online supplementary file 1) is based on the questionnaire used in the DEGREE protocol and has been used in DEGREE-related studies in a number of settings. The baseline core questionnaire represents a minimum dataset and it will provide basic information on exposures such as sociodemographic factors, occupational and environmental exposure, lifestyle, diagnosis of infectious diseases and medication. Local research teams may decide to add data items of specific interest to the core dataset, particularly items of relevance to societal and occupational context and/or environmental samples. They also have the responsibility to translate, validate and to make any local contextual changes. Training procedures for the field staff should be documented.

Researchers will return to field (at least) annually for in-person follow-up visits. At these follow-up visits, participants are invited to respond to a follow-up questionnaire (online supplementary file 2), provide biosamples and update their contact information.

## Clinical measurements

Blood pressure and heart rate should be measured on the right arm after 5 min rest in the sitting position using an automated sphygmomanometer, WHO validated for the clinical setting (eg, Omron HEM-907XL sphygmomanometer), and the average of the first, second and third of three readings is recorded. Subjects height and weight (in meters and kilograms) should be measured (without shoes) using a stadiometer and digital calibrated scales.

## Biosamples

Fasting blood and urine samples will be collected at each study visit and stored in the field into coolers with icebox (4°C) no more than 4 hours before processing.

Dipstick urinalysis should be performed by using electronic readers (urine chemistry analyser) where possible, or otherwise at least 10% of tests should be re-analysed by a second investigator. Parameters that should be reported are urinary specific gravity, pH, protein, blood, leucocytes, nitrite, glucose, etcetera. Investigators with access to ACR measurements may wish to perform these assays (at least at baseline and annually).

Samples for serum analysis should be centrifuged at 3500 rpm for 10 min within 4 hours of collection, and subsequently separated into at least four aliquots of 1–2 mL and stored at ≤−20°C (ideally −80°C). One aliquot should be used for contemporary sCr measurements, for example, by using the modified Jaffe assay (ideally also using standards traceable to isotope dilution mass spectrometry (IDMS) reference material). At baseline and during each study visit, a cross-checking of local lab quality control is highly recommended to ensure that sCr

determinations are comparable as these lab results may guide referral to clinical care for participants during the follow-up period. A further aliquot should be stored for a repeat batch measurement of sCr in all samples (a subset of samples from each study visit will be adequate if IDMS referenced methods are used on initial measurement) at the end of follow-up using a method traceable to an IDMS reference material (and potentially also cystatin C).

The CO-DEGREE group suggests the storage of at least a further two 1–2 mL aliquots of serum and a similar amount of urine in addition to those described above. Additional samples and analyses should be pursued depending on the priorities of the local research team. All samples for future analysis should be stored at ≤−20°C (ideally −80°C) in a local or international biobank. Such a biobank requires an uninterruptible power supply to protect the samples.

Investigators should assess (as part of their public engagement efforts) and, if appropriate, obtain consent from participants for future use of samples for further (specific and/or more general) use both locally and internationally (eg, through the DEGREE collaboration) as well as ensure that storage capacity is available.

### Data management and reporting

Questionnaires and samples will be labelled using a unique bar-code to maintain participant confidentiality. Electronic data capture systems such as Open Data Kit[39] may be the most resource efficient method to capture questionnaire data but where hard copies are used, double data-entry should be undertaken to minimise the transcription errors.

The CO-DEGREE protocols are openly available to interested research teams. Although primarily designed to be used in population-based studies, similar approaches could also be used in an occupational or other selected cohorts.

Each centre will be the 'owner' of their data and expected to publish the results of their study independently. However, where a study is registered as part of the DEGREE collaboration, the coordinating centre will request a digital copy of anonymised individual-level data to allow the undertaking of international comparisons. In addition, a summary of local contextual information and a description of the population characteristics along with response rates will be requested. The importance of such information is emphasised.

### Sample size and follow-up duration

The overall size of the cohort will be largely dependent on the proportion of the 'healthy' population which is expected to experience a 'substantial' decline in eGFR over time in the community as a result of CKDu. As discussed above, demonstrating that reduced renal function without diabetes, hypertension or known kidney diseases is prevalent on a cross-sectional basis is a necessary first step before pursuing this work. If, for example, this study protocol was to be conducted in a general

population sample in Europe or the USA with similar exclusion criteria, there would be very little or no decline of kidney function in the young adult population. In contrast, in our Nicaragua study of apparently healthy adults aged 18–30 years,[25 26] there was a clearly distinct subgroup which experienced a marked decline in kidney function over a short time, whereas the eGFR in the other study participants was relatively stable. Given this distribution of such eGFR trajectories in the population, we would expect any analysis of risk factors to be conducted using a prospective case-control approach.

Therefore, the sample size requirements to detect an association with an exposure at any given power will be determined by the following factors:
1. Proportion of the population that experience 'substantial' decline. In turn the power to detect 'substantial' decline will depend on the following:
   a. The rate of eGFR decline in those affected.
   b. The duration of follow-up.
   c. The number of eGFR measures.
2. Proportion of general population exposed to any exposure of interest.
3. Effect size of any exposure.
4. The study retention rate.

Taking a simplistic approach, the duration of the study should be designed so that those affected have sustained a clinically important loss of kidney function, for example, 20% of normal eGFR. Therefore, if CKDu in the study population is predicted, from a baseline of ≥60 mL/min eGFR, to lead to a loss of eGFR of a magnitude of 5% each year (~7 mL/min/1.73 m$^2$/year), the study duration should be 4 years. If, alternatively, loss is predicted to be 10% each year, study duration could be as short as 2 years. Additional eGFR measures, over and above the suggested annual frequency, will reduce error associated with determining trajectory (and might be performed for the reasons discussed above) but either way a minimum follow-up of 2 years is recommended.

After basing the study duration on the expected rate of eGFR decline among those affected, the sample size can then be calculated on the basis of the expected frequency of 'substantial' decline among the population and the effect size of any proposed exposure that it is desirable to detect. A number of scenarios are outlined in table 1. A further (eg, 20%, depending on local circumstances) increase in target recruitment is advised to allow for loss to follow-up.

Finally, these initial sample sizes will need adjustment for exclusions based on estimates of the prevalence of previously unknown CKD (based on eGFR/albuminuria tests), diabetes, hypertension or other known causes of CKD at baseline (unless these data are already available from a previously conducted cross-sectional study). It is worth considering whether people who may have CKD (or CKD risk factors) will be aware of this, as this may affect the numbers of participants that will be retained for the analysis following testing. For example, if there is screening for kidney problems (as in some Central

American Sugarcane mills or community-based screening in Sri Lanka), then potential cohort participants may be aware of their kidney function status and can be excluded from the study sample prior to recruitment. For example, 5% of the target population in the community studied in Nicaragua reported pre-existing CKD. Nevertheless, there was an additional 10% who had undiagnosed impaired kidney function at baseline assessment based on their laboratory findings, highlighting the importance of identifying an age group where CKDu is not already highly prevalent so as to satisfy a key inclusion criterion (absence of CKD at baseline) when calculating sample sizes.

## Ethics and dissemination

Local research teams will ensure these studies are conducted in accordance with the Declaration of Helsinki Principles and be responsible for assuring that the work is approved by the local institutional review board. Written informed consent will be obtained from all participants before taking part in the study. Information should be transparent in terms of using the data and biosamples stored for future research. Typically, a key aspect of the ethical review of any protocol is a discussion surrounding the provision of feedback and advice to participants when abnormal results become available. In most settings, these processes should be developed in partnership with local communities. Furthermore, mechanisms will need to be established in collaboration with local health providers/ healthcare systems to define pathways for participants needing referral for medical care. Findings from these studies should be disseminated widely by publication in peer-reviewed journals and presentations/representations to relevant local stake holders.

## Patient and public involvement

Patients or member of the public were not involved in the design of this protocol. Procedures will vary by location; however, the DEGREE Steering Committee would encourage active involvement of lay members of study communities in additional design elements and implementation of these studies, particularly relating to the ethical issues above. For example, it is expected that study participants will receive the results of their lab tests, explanations of them and a reference to the relevant health centre, if appropriate. However, the best mechanisms for doing this will vary by location.

## Experience with the CO-DEGREE protocol in Nicaragua

The protocol presented here is, by necessity, generic. The approaches and challenges of implementing the protocol will vary widely in different populations and regions of the world. However, since we have already implemented this protocol in a study in Nicaragua,[25 26] we will make some observations on the practicalities, and challenges, or implementing the protocol in this context.

The Nicaragua study involved community-based follow-up in Leon and Chinandega departments.[25] A number of strategies were used to maximise response and retention rates. As the workday starts very early in the morning and finishes late in the afternoon, attempts were made to conduct data collection during economically less active (eg, each side of the main sugar harvest) periods of the year, so as to still capture approximately 30% of participants who were employed at the time. Additionally, participants receive their kidney test results within a fortnight of the study visits and receive reimbursement of expenses and any lost income they have incurred to attend the study visit. Although study visits have been timetabled to occur outside of the harvest season, employees still express the concern that their employment opportunities might be affected by taking part in the study. In an attempt to mitigate against these types of consequences, the study team have corresponded with local employers explaining the content and extent of this study in order to reduce any concerns about workers' participation. In addition, the study team takes particular precautions to maintain participant's confidentiality during the study and beyond.

Conducting a follow-up study in a rural area remains a major challenge. Alongside the logistical challenges of reaching geographically isolated neighbourhoods along poor-quality roads, a significant obstacle has been internal and external migration due to lack of employment source or social unrest. Rural communities have a tradition of working with seasonal crops, and sugarcane workers often leave their communities at the end of each harvest season, to go abroad or to other regions within the country in search of temporary employment. In our study, at the end of each harvest, up to 30% of the study population had left their communities in search of alternative employment during the non-harvest period in our study. Despite these problems, our team achieved attendance at 92% of all scheduled visits over 2 years.[25 26] However, the level of investment of time and resources should not be underestimated.

Finally, continuing community engagement and the maintenance of good relationships between researchers, community leaders, participants and communication with local healthcare system have been key. The development of standardised procedures for use by the research team may be useful in this context, for example, a reference flowchart for communication with local health posts/ primary hospital or hospital for persons with health problems detected during the study.

## DISCUSSION

The CO-DEGREE protocol was developed in response to the highly prevalent form of CKD of unknown cause that is affecting Mesoamerica and other countries around the globe. To date, the existing epidemiological studies of CKDu have provided an incomplete understanding of the evolution of and risk factors for disease. This CO-DEGREE protocol aims to provide a framework to address this.

This CO-DEGREE protocol is designed to capture the entire at-risk population by aiming to recruit men and women, and those who work across a variety of different occupations. The main outcome measure of within-person loss of eGFR over time, which means it should be possible to capture the earliest stages of disease, making associations with possible causal exposures (and exacerbating factors) less prone to reverse causation and recall bias.

We do not underestimate the challenges posed by the lack of language-validated and standardised exposure questionnaires in this area. The accompanying questionnaire represents a minimum, and most studies will use an expanded dataset. Currently, there is an absence of globally generalisable instrument to capture environmental and occupational exposures; however, the DEGREE group is undertaking further work in this area. Additionally, short-term or long-term environmental measurements and/or novel biomarkers that capture exposure to heat, agrichemicals and/or infection in either the community or workplace are likely to be valuable additions to this type of study but are beyond the scope of this basic protocol.

Finally, it should be emphasised that this protocol is not suitable for studying the progression of CKD in general, due to the specific constraints introduced by excluding those with hypertension, diabetes and CKD as well as other known causes of CKD (ie, those with proteinuria and/or with reduced eGFR) at baseline. Indeed, in settings where there is not a high prevalence of CKDu, a cohort comprising people without traditional risk factors for CKD or with CKD would be unlikely to identify any detectable kidney function loss over time in the young–adult population. For studies outside the CKDu arena, investigators are advised to use alternative methodologies using established protocols, for example, the CRIC study.[40]

In conclusion, we have designed a CO-DEGREE protocol that can be used in the different settings around the globe to investigate the evolution of CKDu and the associated risk factors for decline in kidney function. These studies should provide important information on the early decline in kidney function across different affected areas as well as key insight into the cause(s) of disease.

**Author affiliations**
[1]Research Centre on Health, Work and Environment (CISTA), National Autonomous University of Nicaragua, León, Nicaragua
[2]Centre for Nephrology, University College London, London, UK
[3]Department of Non-Communicable Disease Epidemiology, London School of Hygiene and Tropical Medicine, London, UK
[4]School of Public Health, Faculty of Medicine, Imperial College London, London, UK
[5]Campus Mar, Instituto de Salud Global Barcelona, Barcelona, Spain
[6]CIBER Epidemiología y Salud Pública (CIBERESP), Madrid, Spain
[7]Universitat Pompeu Fabra (UPF), Barcelona, Spain
[8]Division of Nephrology, Department of Internal Medicine & Epidemiology, University of Michigan, Ann Arbor, Michigan, USA
[9]University of Michigan, Ann Arbor, Michigan, USA
[10]La Isla Network, Washington DC, UK
[11]Department of Nephrology and Mineral Metabolism, National Medical Science and Nutrition Institute Salvador Zubirán, Mexico City, Mexico
[12]Department of Public Health and Community Medicine, Insitute of Medicine, Sahlgrenska Academy, University of Gothenburg, Gothenburg, Sweden
[13]Occupational and Environmental Medicine, Sahlgrenska University Hospital, Region Västra Götaland, Gothenburg, Sweden
[14]Brigham and Women's Hospital and Harvard Medical School, Boston, Massachusetts, USA
[15]World Health Organization Country Office, Colombo, Sri Lanka
[16]Division of Nepohrology UBC, University of British Columbia, Vancouver, British Columbia, Canada
[17]Instituto di Ricerche Farmacologiche Mario Negri - IRCCS, Milan, Italy
[18]Deparament of Medical Statistics and Non-Communicable Disease Epidemiology, London School of Hygiene and Tropical Medicine, London, UK
[19]Centre for Global NCDs, London School of Hygiene and Tropical Medicine, London, UK

**Twitter** Marvin Gonzalez-Quiroz @@MarvinGonzlez16

**Acknowledgements** The authors wish to thank all DEGREE Steering Committee members.

**Collaborators** DEGREE Steering Committee members: Neil Pearce (UK) (Chair), Ben Caplin (UK) (Co-chair), Jason Glaser (USA), Ricardo Correa-Rotter (Mexico), Kristina Jakobsson (Sweden), Ajay Singh (USA/India), Antonio Bernabe-Ortiz (Peru), Emmanuel Burdmann (Brazil), Marvin Gonzalez-Quiroz (Nicaragua), Vivekanand Jha (India), Rick Johnson (USA), Phabdheep Kaur (India), Pronpimolk Kongtip (Thailand), Hans Kromhout (Netherlands), Adeera Levin (Canada), Magdalena Madero Rovalo (Mexico), Dorothea Nitsch (UK), Moffat Nyirenda (Ugand/Malawi), Cristina O'Callaghan-Gordo (Spain), Pablo Perel (UK/Argentina), Dorairaj Prabhkaran (India), Narayan Prasad (India), Giuseppe Remuzzi (Italy), Rajiv Saran (USA), Liam Smeeth (UK), Vidhya Venugopal (India).

**Contributors** The CO-DEGREE protocol was conceived by MGQ, BC, DN and NP. Design of the study and drafting the protocol was done by MGQ, BC, DN, NP, SH, COG, RS, JG, RCR, KJ, AS, NG, AL and GR. All authors contributed to and approved the final manuscript.

**Funding** This work was supported by grants from the UK Colt Foundation and the UK Medical Research Council (MR/P02386X/1).

**Competing interests** None declared.

**Patient consent for publication** Not required.

**Provenance and peer review** Not commissioned; externally peer reviewed.

**Data availability statement** Data are available in a public, open access repository. There are no data in this work. Data are available upon reasonable request. Data may be obtained from a third party and are not publicly available. No data are available. All data relevant to the study are included in the article or uploaded as supplementary information.

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
