## [Reviewer comments · BMJ Open]

ARTICLE DETAILS

TITLE (PROVISIONAL)	Rationale and Population-based prospective cohort protocol for the Disadvantaged Populations at Risk of Decline in eGFR (CO-DEGREE)
AUTHORS	Gonzalez-Quiroz, Marvin; Nitsch, Dorothea; Hamilton, Sophie; O'Callaghan Gordo, Cristina; Saran, Rajiv; Glaser, Jason; Correa-Rotter, Ricardo; Jakobsson, Kristina; Singh, Ajay; Gunawardena, Nalika; Levin, Adeera; Remuzzi, Giuseppe; Caplin, Ben; Pearce, Neil

VERSION 1 – REVIEW

REVIEWER	steven rosansky dorn research institute va columbia sc use
REVIEW RETURNED	14-May-2019

GENERAL COMMENTS	this is an ambitious and worthwhile study my main concern is lack of a preliminary hypothesis for the etiology of this CKD it would be of interest to offer urine dipstick tests at the work place, for specific gravity to look for the level of urine concentration, this is one of the hypothesis, prolonged dehydration? without this kind of casual link to test i am not sure what this descriptive study will offer i guess it may tell if there really is a unique CKD and its incidence which is of some value
--

REVIEWER	Jennifer Bragg-Gresham University of Michigan, USA
REVIEW RETURNED	16-May-2019

GENERAL COMMENTS	The topic of CKDu is very important and having a standardized protocol for study in this area will be invaluable. The authors have laid out the specifics needed with special attention to limitations that should be considered based on the specific population of study. I especially appreciated the experience shared with the study already underway in Nicaragua. I only have a few questions/concerns: 1. Why do the authors suggest such a high cut-off for ACR or urine dipstick under the exclusion criteria? Risk of progression as been seen at levels as low as 10 mg/g, while the standard definition for
--

	albuminuria is > 30 mg/g. Using exclusion criteria as high as > 300 mg/g I think may allow individuals into the study who do already have established kidney disease. 2. It may be helpful, although not meant to be specific to the protocol, but a list of potential environmental factors that researchers should be collecting.
--	---

VERSION 1 – AUTHOR RESPONSE

Referee	Comments from reviewer	Response to comments	Location of change in manuscript
Reviewer 1	1. my main concern is lack of a preliminary hypothesis for the etiology of this CKD. it would be of interest to offer urine dipstick tests at the work place, for specific gravity to look for the level of urine concentration, this is one of the hypothesis, prolonged dehydration? without this kind of casual link to test i am not sure what this descriptive study will offer. I guess it may tell if there really is a unique CKD and its incidence which is of some value	Thank you for raising this point. As noted, several plausible hypotheses have been suggested. We have presented a generic study protocol enabling. These hypotheses to be investigated in a variety of settings. Nevertheless, in the bio-samples section, we mentioned that dipstick urinalysis should be performed by using electronic readers and the parameters that should be reported are: urinary specific gravity, pH, protein, blood, etc.	Biosamples Page 16
Reviewer 2	5. Why do the authors suggest such a high cut-off for ACR or urine dipstick under the exclusion criteria? Risk of progression as been seen at levels as low as 10 mg/g, while the standard definition for albuminuria is > 30 mg/g. Using exclusion criteria as high as > 300 mg/g I think may allow individuals into the study who do already have established kidney disease.	The exclusion criteria are designed to exclude significant renal disease (CKD or CKDu) at baseline (or known risk factors for CKD). a cut-off of ACR>300 mg/g is designed to exclude significant glomerular disease. However it is recognized that both the eGFR (<60mL/min) and ACR exclusions will still mean some participants with underlying renal abnormalities are included however we feel this	Population and sampling strategy Page 14

		is a pragmatic approach to excluding those with either established CKDu or those unable to develop it (because of risk factors for or existing evidence of other forms of CKD). We have added a sentence to clarify this.	
	2. It may be helpful, although not meant to be specific to the protocol, but a list of potential environmental factors that researchers should be collecting.	Environmental samples will usually be collected, but this will depend on the local context and capacity, and therefore was not discussed in detail here. In addition, the questionnaire will provide information on various exposures such as: sociodemographic factors, occupational and environmental exposure, lifestyle, diagnosis of infectious diseases, and medication	Questionnaire Page 15